# Uncertainty-aware Self-training for Few-shot Text Classification

**Subhabrata Mukherjee**
Microsoft Research
Redmond, WA
submukhe@microsoft.com

**Ahmed Hassan Awadallah**
Microsoft Research
Redmond, WA
hassanam@microsoft.com

## Abstract

Recent success of pre-trained language models crucially hinges on fine-tuning them on large amounts of labeled data for the downstream task, that are typically expensive to acquire or difficult to access for many applications. We study self-training as one of the earliest semi-supervised learning approaches to reduce the annotation bottleneck by making use of large-scale unlabeled data for the target task. Standard self-training mechanism randomly samples instances from the unlabeled pool to generate pseudo-labels and augment labeled data. We propose an approach to improve self-training by incorporating uncertainty estimates of the underlying neural network leveraging recent advances in Bayesian deep learning. Specifically, we propose (i) acquisition functions to select instances from the unlabeled pool leveraging Monte Carlo (MC) Dropout, and (ii) learning mechanism leveraging model confidence for self-training. As an application, we focus on text classification with five benchmark datasets. We show our methods leveraging only 20-30 labeled samples per class for each task for training and for validation perform within 3% of fully supervised pre-trained language models fine-tuned on thousands of labels with an aggregate accuracy of 91% and improvement of up to 12% over baselines.

## 1   Introduction

**Motivation.** Deep neural networks are the state-of-the-art for various applications. However, one of the biggest challenges facing them is the lack of labeled data to train these complex networks. Not only is acquiring large amounts of labeled data for every task expensive and time consuming, but also it is not feasible to perform large-scale human labeling, in many cases, due to data access and privacy constraints. Recent advances in pre-training help close this gap. In this, deep and large neural networks like BERT [Devlin et al., 2019], GPT-2 [Radford et al., 2019] and RoBERTa [Liu et al., 2019] are trained on millions of documents in a self-supervised fashion to obtain general purpose language representations. However, even with a pre-trained model, we still need task-specific fine-tuning that typically requires thousands of labeled instances to reach state-of-the-art performance. For instance, our experiments show $16\%$ relative improvement when fine-tuning BERT with the full training set ($25K$-$560K$ labels) vs. fine-tuning with only 30 labels per class. Recent work [Wang et al., 2020a] show this gap to be bigger for structured learning tasks such as sequence labeling.

Semi-supervised learning (SSL) [Chapelle et al., 2010] is one of the promising paradigms to address this shortcoming by making effective use of large amounts of unlabeled data in addition to some labeled data for task-specific fine-tuning. Recent work [Xie et al., 2019] on leveraging SSL with consistency learning has shown state-of-the-art performance for text classification with limited labels leveraging auxiliary resources like back-translation and forms a strong baseline for our work.

Self-training (ST, [Scudder, 1965]) as one of the earliest SSL approaches has recently been shown to obtain state-of-the-art performance for tasks like neural machine translation [He et al., 2019], named

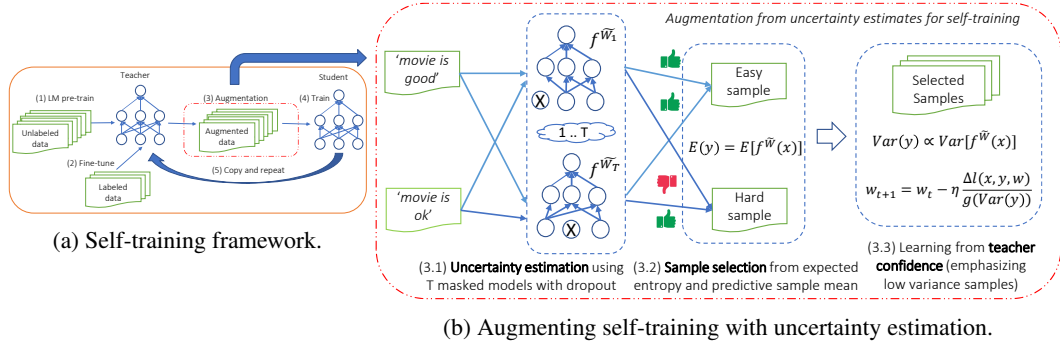

(a) Self-training framework.

(b) Augmenting self-training with uncertainty estimation.

Figure 1: Uncertainty-aware self-training framework.

entity recognition and slot tagging for task-oriented dialog systems [Wang et al., 2020a]; performing at par with supervised systems without using any auxiliary resources. For self-training, a base model (*teacher*) is trained on some amount of labeled data and used to pseudo-annotate (task-specific) unlabeled data. The original labeled data is augmented with the pseudo-labeled data and used to train a *student* model. The student-teacher training is repeated until convergence. Such frameworks have also been recently used for distillation [Wang et al., 2020b, Mukherjee and Hassan Awadallah, 2020] to transfer knowledge from huge pre-trained language models to shallow student models for efficient inference often operating over task-specific labeled data and unlabeled transfer data.

Traditionally, self-training mechanisms do not consider the teacher uncertainty or perform any sample selection during the pseudo-labeling process. This may result in gradual drifts from self-training on noisy pseudo-labeled instances [Zhang et al., 2017]. Sample selection leveraging teacher confidence has been studied in curriculum learning [Bengio et al., 2009] and self-paced learning [Kumar et al., 2010] frameworks. These works leverage the *easiness* of the samples to inform a learning schedule like training on easy concepts first followed by complex ones. Since it is hard to assess the *easiness* of a sample, especially in deep neural network based architectures, these works rely only on the teacher model loss, while ignoring its uncertainties, for sample selection.

Intuitively, if the teacher model already predicts some samples with high confidence, then there is little to gain with self-training if we focus only on these samples. On the other hand, hard examples for which the teacher model has less confidence are hard to rely on for self-training as these could be noisy or too difficult to learn from. In this scenario, the model could benefit from judiciously selecting examples for which the teacher model is *uncertain* about. However, it is non-trivial to generate uncertainty estimates for non-probabilistic models like deep neural networks. To this end, we leverage recent advances in Bayesian deep learning [Gal and Ghahramani, 2016] to obtain uncertainty estimates of the teacher for pseudo-labeling and improving the self-training process.

**Our task and framework overview.** We focus on leveraging pre-trained language models for classification with few labeled samples (e.g., $K = \{20, 30\}$) per class for training and validation, and large amounts of task-specific unlabeled data. Figure 1(a) shows an overview of a traditional self-training framework, where augmented data is obtained from hard pseudo-labels from the teacher (e.g., BERT [Devlin et al., 2019]) without accounting for its uncertainty. Figure 1(b) shows an overview of our uncertainty-aware self-training framework (UST)[1]. We extend the traditional self-training framework with *three* core components, namely: (i) *Masked model dropout for uncertainty estimation:* We adopt MC dropout [Gal and Ghahramani, 2016] as a technique to obtain uncertainty estimates from the pre-trained language model. In this, we apply stochastic dropouts after different hidden layers in the neural network model and approximate the model output as a random sample from the posterior distribution. This allows us to compute the model uncertainty in terms of the stochastic mean and variance of the samples with a few stochastic forward passes through the network. (ii) *Sample selection.* Given the above uncertainty estimates for a sample, we employ entropy-based measures to select samples that the teacher is most or least confused about to infuse for self-training corresponding to *easy-* and *hard*-entropy-aware example mining. (iii) *Confident learning.* In this, we train the student model to explicitly account for the teacher confidence by emphasizing on the low variance examples. All of the above components are jointly used for end-to-end learning. We adopt BERT as our encoder and show that its performance can be significantly improved by an average of 12% for few-shot settings without using any auxiliary resources. Furthermore, we also

outperform recent models [Xie et al., 2019] that make use of auxiliary resources like back-translation. In summary, our work makes the following contributions. (i) Develops an uncertainty-aware self-training framework for few-shot text classification. (ii) Compares the effectiveness of various sample selection schemes leveraging teacher uncertainty for self-training. (iii) Demonstrates its effectiveness for text classification with few labeled samples on five benchmark datasets.

## 2 Background

Consider $D_l = \{x_i, y_i\}$ to be a set of $n$ labeled instances with $y_i$ being the class label for $x_i$. Each $x_i$ is a sequence of $m$ tokens: $x_i = \{x_{i1}, x_{i2} \cdots x_{im}\}$. Also, consider $D_u = \{x_j\}$ to be a set of $N$ unlabeled instances, where $n \ll N$. For most tasks, we have access to a small amount of labeled data along with a larger amount of unlabeled ones.

**Self-training** starts with a base *teacher* model trained on the labeled set $D_l$. The teacher model is applied to a subset $S_u \subset D_u$ of the unlabeled data $D_u$ to obtain pseudo-labeled instances. The augmented data $D_l \cup S_u$ is used to train a *student* model. The teacher-student training schedules are repeated till a convergence criterion is satisfied. The unlabeled subset $S$ is usually selected based on confidence scores of the teacher model. In Section 3.1, we study different techniques to generate this subset leveraging uncertainty of the teacher model. Self-training process can be formulated as:

$$min_W \; \mathbb{E}_{x_l, y_l \in D_l}[-log \; p(y_l|x_l; W)] + \lambda \mathbb{E}_{x_u \in S_u, S_u \subset D_u} \mathbb{E}_{y \sim p(y|x_u; W^*)}[-log \; p(y|x_u; W)] \quad (1)$$

where $p(y|x; W)$ is the conditional distribution under model parameters $W$. $W^*$ is given by the model parameters from the last iteration and fixed in the current iteration. Similar optimization functions have been used recently in variants of self-training for neural sequence generation [He et al., 2019], data augmentation [Xie et al., 2019] and knowledge distillation.

**Bayesian neural network** (BNN) [Gal and Ghahramani, 2015] assumes a prior distribution over its weights, thereby, replacing a deterministic model's weight parameters by a distribution over these parameters. For inference, instead of directly optimizing for the weights, BNN averages over all the possible weights, also referred to as marginalization.

Consider $f^W(x) \in \mathbb{R}^h$ to be the $h-$dimensional output of such a neural network where the model likelihood is given by $p(y|f^W(x))$. For classification, we can further apply a softmax likelihood to the output to obtain: $P(y = c|x, W) = softmax(f^W(x)).$ \quad (2)

Bayesian inference aims to find the posterior distribution over the model parameters $p(W|X, Y)$. Given an instance $x$, the probability distribution over the classes is given by marginalization over the posterior distribution as: $p(y = c|x) = \int_W p(y = c|f^W(x))p(W|X, Y)dW$.

This requires averaging over all possible model weights, which is intractable in practice. Therefore, several approximation methods have been developed based on variational inference methods and stochastic regularization techniques using dropouts. Here, the objective is to find a surrogate distribution $q_\theta(w)$ in a tractable family of distributions that can replace the true model posterior that is hard to compute. The ideal surrogate is identified by minimizing the Kullback-Leibler (KL) divergence between the candidate and the true posterior.

Consider $q_\theta(W)$ to be the Dropout distribution [Srivastava et al., 2014] which allows us to sample $T$ masked model weights $\{\widetilde{W}_t\}_{t=1}^T \sim q_\theta(W)$. For classification tasks, the approximate posterior can be now obtained by Monte-Carlo integration as:

$$p(y = c|x) \approx p(y = c|f^W(x))q_\theta(W)dW$$
$$\approx \frac{1}{T} \sum_{t=1}^T p(y = c|f^{\widetilde{W}_t}(x)) = \frac{1}{T} \sum_{t=1}^T softmax(f^{\widetilde{W}_t}(x)) \quad (3)$$

## 3 Uncertainty-aware Self-training

Given a pre-trained language model as the teacher, we first fine-tune it on the small amount of labeled data. To this end, we use a *small* batch size to gradually expose the teacher model to the few available labels. Given our low-resource setting, we do not compute uncertainty estimates over the small

labeled set. Instead, given the teacher model, we compute uncertainty estimates over each instance from the large unlabeled set as follows. Considering dropouts enabled before every hidden layer in the teacher model, we perform several stochastic forward passes through the network for every unlabeled sample. For computational efficiency, we perform these stochastic passes and hence the self-training over sampled mini-batches.

For each unlabeled instance $x_u$, given $T$ stochastic forward passes through the network with dropout, each pass $t \in T$ with corresponding model parameters $\widetilde{W_t} \sim q_\theta(W)$, generates a pseudo-label given by Equation (2) as $p(y_t^*) = softmax(f^{\widetilde{W_t}}(x_u))$.

There are several choices to integrate this pseudo-label for self-training, including, considering $E(y) = \frac{1}{T} \sum_{t=1}^{T} softmax(f^{\widetilde{W_t}}(x))$ for the soft pseudo-labels as well as discretizing them for hard labels and aggregating predictions from the $T$ passes as:

$$y_u = argmax_c \sum_{t=1}^{T} \mathbb{I}[argmax_{c'}(p(y_t^* = c')) = c] \qquad (4)$$

where $\mathbb{I}(.)$ is an indicator function. Empirically, the hard pseudo-labels work better in our framework with standard log loss. Similar observation has been reported in contemporary works [Kumar et al., 2020, Wang et al., 2020a] in self-training, which refer to this as *label sharpening*. The pseudo-labeled data is used to augment and re-train the model with the steps repeated until convergence. At each self-training iteration, the model parameters $W^*$ from the previous iteration are used to compute the predictive mean $E(y)$ of the samples before re-training the model end-to-end on the augmented (pseudo-labeled) data to learn the new parameters $W$.

In order to incorporate the above uncertainty measures in the self-training framework, we modify the loss component over unlabeled data in the original self-training learning process (Equation 1) as:

$$min_{W,\theta} \; \mathbb{E}_{x_u \in S_u, S_u \subset D_u} \; \mathbb{E}_{\widetilde{W} \sim q_\theta(W^*)} \; \mathbb{E}_{y \sim p(y|f^{\widetilde{W}}(x_u))}[-log \, p(y|f^W(x_u))] \qquad (5)$$

where $W^*$ denotes the model parameters from the previous iteration of the self-training process.

### 3.1 Sample Selection

Prior works have leveraged various measures to sample instances based on predictive entropy [Shannon, 2001], variation ratios [Freeman, 1965], standard deviation and more recently based on model uncertainty, like Bayesian Active Learning by Disagreement (BALD) [Houlsby et al., 2011] leveraging stochastic dropouts. Consider $D'_u = \{x_u, y_u\}$ to be the pseudo-labeled dataset obtained by applying the teacher model to the unlabeled data. The objective of the BALD measure is to select samples that maximize the information gain about the model parameters, or in other words, maximizing the information gain between predictions and the model posterior given by: $\mathbb{B}(y_u, W|x_u, D'_u) = \mathbb{H}[y_u|x_u, D'_u] - \mathbb{E}_{p(W|D'_u)}[\mathbb{H}[y_u|x_u, W]]$, where $H[y_u|x_u, W]$ denotes the entropy of $y_u$ given $x_u$ under model parameters $W$. Gal et al. [2017] show that the above measure can be approximated with the Dropout distribution $q_\theta(W)$ such that:

$$\widehat{\mathbb{B}}(y_u, W|x_u, D'_u) = - \sum_c \Big(\frac{1}{T} \sum_t \hat{p}_c^t \Big) log \Big(\frac{1}{T} \sum_t \hat{p}_c^t \Big) + \frac{1}{T} \sum_{t,c} \hat{p}_c^t log(\hat{p}_c^t) \qquad (6)$$

where, $\hat{p}_c^t = p(y_u = c|f^{\widetilde{W_t}}(x_u)) = softmax(f^{\widetilde{W_t}}(x_u))$.

The above measure depicts the decrease in the expected posterior entropy in the output space $y$. This results in a tractable estimation of the BALD acquisition function with $\widehat{\mathbb{B}}(y_u, W|.) \xrightarrow[T \to \infty]{}$ $\mathbb{B}(y_u, W|.)$. A high value of $\widehat{\mathbb{B}}(y_u, W|x_u, D'_u)$ indicates that the teacher model is highly confused about the expected label of the instance $x_u$. We use this measure to rank all the unlabeled instances based on uncertainty for further selection for self-training.

**Class-dependent selection.** We can further modify this measure to take into account the expected class label of the instance. This helps in sampling equivalent number of instances per class, and avoids the setting where a particular class is typically hard, and the model mostly samples instances from that class. Given the pseudo-labeled set $S_u$, we can construct the set $\{x_u \in S_{u,c} : y_u = c\}$ for

---
**Algorithm 1:** Uncertainty-aware self-training (UST).
---
Continue pre-training teacher language model on task-specific unlabeled data $D_u$ ;
Fine-tune model $f^W$ with parameters $W$ on task-specific small labeled data $D_l$ ;
**while** *not converged* **do**
    Randomly sample $S_u$ unlabeled examples from $D_u$ ;
    **for** $x \in S_u$ **do**
        **for** $t \leftarrow 1$ **to** $T$ **do**
            $W_t \sim Dropout(W)$ ;
            $y_t^* = softmax(f^{W_t}(x))$;
        **end**
        Compute predictive sample mean $E(y)$ and predictive sample variance $Var(y)$ with Equation 9 ;
        Compute BALD acquisition function with Equation 6 ;
    **end**
    Sample $R$ instances from $S_u$ employing sample selection with Equations 7 or 8 ;
    Pseudo-label $R$ sampled instances with model $f^W$ ;
    Re-train model on $R$ pseudo-labeled instances with Equation 12 and update parameters $W$ ;
**end**
---

every class $c$. Now, we use the BALD measure to select instances from each class-specific set instead of a global selection.

**Selection with exploration.** Given the above measure, there are choices to select the pseudo-labeled examples for self-training, including mining hard ones and easy ones (as in curriculum learning and self-paced learning). To this end, we can select the top-scoring instances for which the model is *least* or *most* uncertain about, ranked by $1 - \widehat{\mathbb{B}}(y_u, W|x_u, D'_u)$ and $\widehat{\mathbb{B}}(y_u, W|x_u, D'_u)$ respectively. In the former case, if the model is always certain about some examples, then these might be too easy to contribute any additional information. In the latter case, emphasizing only on the hard examples may result in drift due to noisy pseudo-labels. Therefore, we want to select examples with some *exploration* to balance these schemes with sampling using the uncertainty masses. To this end, given a budget of $R$ examples to select, we sample instances $x_u \in S_{u,c}$ without replacement with probability:

$$p_{u,c}^{easy} = \frac{1 - \widehat{\mathbb{B}}(y_u, W|x_u, D'_u)}{\sum_{x_u \in S_{u,c}} 1 - \widehat{\mathbb{B}}(y_u, W|x_u, D'_u)} \quad (7) \qquad p_{u,c}^{hard} = \frac{\widehat{\mathbb{B}}(y_u, W|x_u, D'_u)}{\sum_{x_u \in S_{u,c}} \widehat{\mathbb{B}}(y_u, W|x_u, D'_u)} \quad (8)$$

Our framework can use either of the above two strategies for selecting pseudo-labeled samples from the unlabeled pool for self-training; where these strategies bias the sampling process towards picking *easier* samples (less uncertainty) or *harder* ones (more uncertainty) for re-training.

### 3.2  Confident Learning

The above sampling strategies select informative samples for self-training conditioned on the posterior entropy in the label space. However, they use only the predictive mean, while ignoring the uncertainty of the model in terms of the predictive variance. Note that many of these strategies implicitly minimize the model variance (e.g., by focusing more on difficult examples for hard example mining). The prediction uncertainty of the teacher model is given by the variance of the marginal distribution, where the overall variance can be computed as:

$$Var(y) = Var[\mathbb{E}(y|W, x)] + \mathbb{E}[Var(y|W, x)] \tag{9}$$

$$= Var(softmax(f^W(x)) + \sigma^2 \tag{10}$$

$$\approx \left( \frac{1}{T} \sum_{t=1}^{T} y_t^{*}(x)^T y_t^{*}(x) - E(y)^T E(y) \right) + \sigma^2 \tag{11}$$

where, $y_t^{*}(x) = softmax(f^{\widetilde{W_t}}(x))$ and the predictive mean computed as: $E(y) = \frac{1}{T} \sum_{t=1}^{T} y_t^{*}(x)$.

We observe the total variance can be decomposed as a linear combination of the model uncertainty from parameters $W$ and the second component results from noise in the data generation process.

In this phase, we want to train the student model to explicitly account for the teacher uncertainty for the pseudo-labels in terms of their predictive variance. This allows the student model to selectively focus more on the pseudo-labeled samples that the teacher is more confident on (corresponding to low variance samples) compared to the less certain ones (corresponding to high variance ones). Accordingly, we update the loss function over the unlabeled data in the self-training mechanism given by Equation 5 to update the student model parameters as:

$$min_{W,\theta} \; \mathbb{E}_{x_u \in S_u, S_u \subset D_u} \; \mathbb{E}_{\widetilde{W} \sim q_\theta(W^*)} \; \mathbb{E}_{y \sim p(y|f^{\widetilde{W}}(x_u))}[log \; p(y|f^W(x_u)) \cdot log \; Var(y)] \qquad (12)$$

In the above equation, the per-sample loss for an instance $x_u$ is a combination of the log loss $-log \; p(y)$ and (inverse of) its predictive variance given by $log \; \frac{1}{Var(y)}$ with $log$ transformation for scaling. This penalizes the student model more on mis-classifying instances that the teacher is more certain on (i.e. low variance samples), and vice-versa.

**Implementation details.** Algorithm 1 outlines the uncertainty-aware self-training process. In our experiments, we employ a single model for self-training. Essentially, we copy teacher model parameters to use as the student model and continue self-training. Although, some works re-initialize the student model from scratch. *Sample size.* Ideally, we need to perform $T$ stochastic forward passes for each sample in the large unlabeled pool which is quite slow for all practical purposes. Therefore, for computational efficiency, at each self-training iteration, we randomly select $S_u$ samples from the unlabeled set, and then select $R \in S_u$ samples from therein based on uncertainty estimates using several stochastic forward passes.

# 4 Experiments

**Encoder.** Pre-trained language models like BERT [Devlin et al., 2019], GPT-2 [Radford et al., 2019] and RoBERTa [Liu et al., 2019] have shown state-of-the-art performance for various natural language processing tasks. In this work we adopt one of these namely, BERT as our base encoder or teacher model. We initialize the teacher model with the publicly available pre-trained checkpoint [Devlin et al., 2019]. To adapt the teacher language model for every downstream task, we further continue pre-training on task-specific unlabeled data $D_u$ using the original language modeling objective. The teacher is finally fine-tuned on task-specific labeled data $D_l$ to give us the base model for self-training.

**Datasets.** We perform large-scale experiments with data from five domains for different tasks as summarized in Table 1. SST-2 [Socher et al., 2013], IMDB [Maas et al., 2011] and Elec [McAuley and Leskovec, 2013] are used for sentiment classification for movie reviews and Amazon electronics product reviews respectively. The other two datasets Dbpedia [Zhang et al., 2015] and Ag News [Zhang et al., 2015]

Table 1: Dataset summary (W: avg. words / doc).

| Dataset | Class | Train | Test | Unlabeled | #W |
|---|---|---|---|---|---|
| IMDB | 2 | 25K | 25K | 50K | 235 |
| DBpedia | 14 | 560K | 70K | - | 51 |
| AG News | 4 | 120K | 7.6K | - | 40 |
| Elec | 2 | 25K | 25K | 200K | 108 |

are used for topic classification of Wikipedia and news articles respectively. For every dataset, we sample $K$ labeled instances from Train data, and add remaining to the Unlabeled data in Table 1.

**Evaluation setting.** For self-training, we fine-tune the base model (teacher) on $K$ labeled instances for each task to start with. Specifically, we consider $K = 30$ instances for each class for training and similar for validation, that are randomly sampled from the corresponding Train data in Table 1. We also show results of the final model on varying $K \in \{20, 30, 50, 100, 500, 1000\}$. We repeat each experiment five times with different random seeds and data splits, use the validation split to select the best model, and report the mean accuracy on the blind test data. We implement our framework in Tensorflow and use four Tesla V100 GPUs for experimentation. We use Adam [Kingma and Ba, 2015] as the optimizer with early stopping and use the best model found so far from the validation loss for all the models. Hyper-parameter configurations with detailed model settings presented in Appendix. We report results from our UST framework with easy sample selection strategy employing Equation 7, unless otherwise mentioned.

**Baselines**. Our first baseline is BERT-Base with 110 MM parameters fine-tuned on $K$ labeled samples $D_l$ for downstream tasks with a small batch-size of $4$ samples, and remaining hyper-parameters retained from its original implementation. Our second baseline, is a recent work UDA [Xie et al.,

Table 2: Accuracy comparison of different models for text classification on five benchmark datasets. All models use the same BERT-Base encoder and pre-training mechanism. All models (except 'all train') are trained with 30 labeled samples for each class and overall accuracy aggregated over five different runs with different random seeds. The accuracy number for each task is followed by the standard deviation in parentheses and percentage improvement ($\uparrow$) over few-shot BERT.

| Dataset | All train | | | | |
|---|---|---|---|---|---|
| | BERT | BERT | UDA | Classic ST | UST (our method) |
| SST | 92.12 | 69.79 (6.45) | 83.58 (2.64) ($\uparrow$ 19.8) | 84.81 (1.99) ($\uparrow$ 21.5) | **88.19** (1.01) ($\uparrow$ 26.4) |
| IMDB | 91.70 | 73.03 (6.94) | **89.30** (2.05) ($\uparrow$ 22.3) | 78.97 (8.52) ($\uparrow$ 8.1) | 89.21 (0.83) ($\uparrow$ 22.2) |
| Elec | 93.46 | 82.92 (3.34) | 89.64 (2.13) ($\uparrow$ 8.1) | 89.92 (0.36) ($\uparrow$ 8.4) | **91.27** (0.31) ($\uparrow$ 10.1) |
| AG News | 92.12 | 80.74 (3.65) | 85.92 (0.71) ($\uparrow$ 6.4) | 84.62 (4.81) ($\uparrow$ 4.8) | **87.74** (0.54) ($\uparrow$ 8.7) |
| DbPedia | 99.26 | 97.77 (0.40) | 96.88 (0.58) ($\downarrow$ 0.9) | 98.39 (0.64) ($\uparrow$ 0.6) | **98.57** (0.18) ($\uparrow$ 0.8) |
| Average | 93.73 | 80.85 (4.16) | 89.06 (1.62) ($\uparrow$ 10.2) | 87.34 (3.26) ($\uparrow$ 8.0) | **91.00** (0.57) ($\uparrow$ 12.6) |

*The header "30 labels per class for training and for validation" spans the last four columns.*

2019] leveraging *back-translation*[2] for data augmentation for text classification. UDA follows similar principles as Virtual Adversarial Training (VAT) [Miyato et al., 2017] and consistency training [Laine and Aila, 2017, Sajjadi et al., 2016] such that the model prediction for the original instance is similar to that for the augmented instance with a small perturbation. In contrast to prior works for image augmentation (e.g., flipping and cropping), UDA leverages back-translation for text augmentation. In contrast to other baselines, this requires auxiliary resources in terms of a trained NMT system to generate the back-translation. Our third baseline is the standard self-training mechanism without any uncertainty. In this, we train the teacher model on $D_l$ to generate pseudo-labels on $D_u$, train the student model on pseudo-labeled and augmented data, and repeat the teacher-student training till convergence. Finally, we also compare against prior SSL works – employing semi-supervised sequence learning [Dai and Le, 2015], adversarial training [Goodfellow et al., 2015, Miyato et al., 2017], variational pre-training [Gururangan et al., 2019], reinforcement learning [Li and Ye, 2018], temporal ensembling and mean teacher models [Laine and Aila, 2017, Tarvainen and Valpola, 2017, Sajjadi et al., 2016], layer partitioning [Li and Sethy, 2019] and delta training [Jo and Cinarel, 2019] – on these benchmark datasets on the same Test data and report numbers from corresponding works.

**Overall comparison.** Table 2 shows a comparison between the different methods. We observe that the base teacher model trained with only 30 labeled samples for each class for each task has a reasonable good performance with an aggregate accuracy of $80.85\%$. This largely stems from using BERT as the encoder starting from a pre-trained checkpoint instead of a randomly initialized encoder, thereby, demonstrating the effectiveness of pre-trained language models as natural few-shot learners. We observe the classic self-training approach leveraging unlabeled data to improve over the base model by $8\%$. UDA leverages auxiliary resources in the form of back-translation from an NMT system for augmentation to improve by over $10\%$. Finally, our UST method obtains the best performance by improving more than $12\%$ over the base model, $4\%$ over classic ST and $2\%$ over UDA *without* any additional resources. Note that our UDA results are different from the original work due to different sequence length and batch sizes resulting from V100 GPU memory constraints.

Our method reduces the overall model variance in terms of both *implicit* reduction by selecting samples with low uncertainty for self-training and *explicit* reduction by optimizing for the sample variance for confident learning. This is demonstrated in a consistent performance of the model across different runs with an aggregated (least) standard deviation of $0.57$ across different runs of the model for different tasks with different random seeds. UDA with its consistency learning closely follows suit with an aggregated standard deviation of $1.62$ across different runs for different tasks. Classic ST without any such mechanism shows high variance in performance across runs with different seeds. In Table 4, we show the results from other works on these datasets as reported in [Li and Ye, 2018, Jo and Cinarel, 2019, Li and Sethy, 2019, Gururangan et al., 2019][3]. We observe our model to obtain at least $7\%$ improvement in IMDB and $4\%$ improvement in AG News over our closest baseline in the

Table 3: Ablation analysis of our framework with different sample selection strategies and components including class-dependent sample selection with exploration (*Class*) and confident learning (*Conf*) for uncertainty-aware self-training with 30 labeled examples per class for training and for validation.

|  | SST | IMDB | Elec | AG News | Dbpedia | Average |
|---|---|---|---|---|---|---|
| BERT | 69.79 | 73.03 | 82.92 | 80.74 | 97.77 | 80.85 |
| Classic ST (Uniform) | 84.81 | 78.97 | 89.92 | 84.62 | 98.39 | 87.34 |
| UST (Easy) | 88.19 | 89.21 | **91.27** | **87.74** | **98.57** | **91.00** |
| - removing *Class* | 87.33 | 87.22 | 89.18 | 86.88 | 98.27 | 89.78 |
| - removing *Conf* | 86.73 | **90.00** | 90.40 | 84.17 | 98.49 | 89.96 |
| UST (Hard) | 88.02 | 88.49 | 90.00 | 85.02 | 98.56 | 90.02 |
| - removing *Class* | 80.45 | 89.28 | 90.07 | 83.07 | 98.46 | 88.27 |
| - removing *Conf* | **88.48** | 87.93 | 88.74 | 84.45 | 98.26 | 89.57 |

form of variational pre-training [Gururangan et al., 2019] and reinforcement learning with adverarial training [Li and Ye, 2018], while using $3x$-$6x$ less training labels (shown by $K$ in Table 4).

**Ablation analysis.** We compare the impact of different components of our model for self-training with 30 labeled examples per class for each task for training and for validation with results in Table 3. *Sampling strategies.* The backbone of the sample selection method in our self-training framework is given by the BALD measure [Houlsby et al., 2011] that has been shown to outperform other active sampling strategies leveraging measures like entropy and variation ratios in Gal et al. [2017] for image classification. We use this measure in our framework to sample examples based on whether the model is confused about the example or not by leveraging sampling strategies in Equations 8 or 7 and optimized by self-training with Equation 12 – denoted by *UST (Hard)* and *UST (Easy)* respectively in Table 3. In contrast to works in active learning that find hard examples to be more informative than easy ones for manual labeling, in the self-training framework we observe the opposite, where hard examples often contribute noisy pseudo-labels. We compare this with uniform sampling in the classic ST framework, and observe that sample selection bias (easy or hard) benefits self-training.

*Class-dependent selection with exploration.* In this, we remove the class-dependent selection and exploration with global selection of samples based on their easiness or hardness for the corresponding UST sampling strategy. Class-dependent selection ameliorates model bias towards picking samples from a specific class that might be too easy or hard to learn from with balanced selection of samples across all the classes, and improves our model on aggregate.

*Confident learning.* In this, we remove confident learning from the UST framework. Therefore, we optimize the unlabeled data loss for self-training using Equation 5 instead of Equation 12 that is used in all other UST strategies. This component helps the student to focus more on examples the teacher is confident about corresponding to low-variance ones, and improves the model on aggregate.

Overall, we observe that each of the above uncertainty-based sample selection and learning strategies outperform the classic self-training mechanism selecting samples uniform at random.

**Impact of $K$ labeled examples.** In Figure 2, we fix the random seed and vary the training labels. We observe the self-training accuracy to gradually improve with increase in the number of labeled examples per class to train the base teacher model leading to better initialization of the self-training process. With only 20 labeled examples for each task for training and for validation, we observe the aggregate performance across five tasks to be $89.27\%$ with further improvements with more labeled data coming from IMDB and AG news datasets. For tasks like DBpedia and Elec with very high performance given few training labels, there is diminishing returns on injecting more labels.

**Impact of self-training iterations.** Figure 3 shows increase in self-training accuracy of UST over iterations for a single run. In general, we observe the self-training performance to improve rapidly initially, and gradually converge in 15-20 iterations. We also observe some models to drift a bit while continuing the self-training process and similar for consistency learning in UDA beyond a certain point. This necessitates the use of the validation set for early termination based on validation loss.

## 5 Related Work

**Semi-supervised learning** has been widely used in many different flavors including consistency training [Bachman et al., 2014, Rasmus et al., 2015, Laine and Aila, 2017, Tarvainen and Valpola, 2017], latent variable models [Kingma et al., 2014] for sentence compression [Miao and Blunsom,

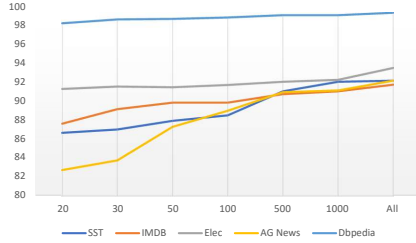

Figure 2: UST accuracy with $K$ train labels/class.

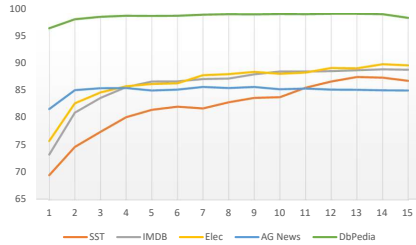

Figure 3: UST accuracy over iterations.

| Datasets | Model | K Labels | Acc. |
|---|---|---|---|
| IMDB | **UST (ours)** | **30** | **89.2** |
| | Variational Pre-training | 200 | 82.2 |
| | Reinforcement + Adv. Training | 100 | 82.1 |
| | SeqSSL + Self-training | 100 | 79.7 |
| | SeqSSL | 100 | 77.4 |
| | Layer Parti. + Temp. Ensembling | 100 | 75.9 |
| | SeqSSL + Adv. Training | 100 | 75.7 |
| | Delta-training | 212 | 75.0 |
| | Layer Parti. + $\Pi$ Model | 100 | 69.3 |
| AG News | **UST (ours)** | **30** | **87.7** |
| | Variational Pre-training | 200 | 83.9 |
| | Reinforcement + Adv. Training | 100 | 81.7 |
| | SeqSSL + Self-training | 100 | 78.5 |
| | SeqSSL | 100 | 76.2 |
| | SeqSSL + Adv. Training | 100 | 73.0 |
| DBpedia | **UST (ours)** | **30** | **98.6** |
| | Reinforcement + Adv. Training | 100 | 98.5 |
| | SeqSSL + Self-training | 100 | 98.1 |
| | SeqSSL + Adv. Training | 100 | 96.7 |
| | SeqSSL | 100 | 96.1 |

Table 4: SSL methods with $K$ train labels/class (Adv: Adversarial, Parti: Partitioning, Temp: Temporal).

2016] and code generation [Yin et al., 2018]. More recently, consistency-based model like UDA [Xie et al., 2019] has shown promising results for few-shot learning for classification leveraging auxiliary resources like paraphrasing and back-translation (BT) [Sennrich et al., 2016].

**Sample selection.** One of the earlier works in neural networks leveraging easiness of the samples for learning is given by curriculum learning [Bengio et al., 2009]. This is based on the idea of learning easier aspects of the task *first* followed by the more complex ones. However, the main challenge is the identification of easy and hard samples in absence of external knowledge. Prior work leveraging self-paced learning [Kumar et al., 2010] and more recently self-paced co-training [Ma et al., 2017] leverage teacher confidence (or lower model loss) to select easy samples during training. In a similar flavor, some recent works have also focused on sample selection for self-training leveraging meta-learning [Li et al., 2019] and active learning [Panagiota Mastoropoulou, 2019, Chang et al., 2017] based on teacher confidence. However, all of these techniques rely on only the teacher confidence while ignoring the uncertainty associated with its predictions. In a recent extension of this work to sequence labeling for named entity recognition and slot tagging for task-oriented dialog systems, Wang et al. [2020a] leverage meta-learning for adaptive sample re-weighting to mitigate error propagation from noisy pseudo-labels. There are also works on anti-curriculum learning (or hard example mining) [Shrivastava et al., 2016] that leverage hardness of the samples.

**Uncertainty in neural networks.** A principled mechanism to generate uncertainty estimates is provided by Bayesian frameworks. A Bayesian neural network Gal and Ghahramani [2016] replaces a deterministic model's weight parameters with distributions over model parameters. Parameter optimization is replaced by marginalisation over all possible weights. It is difficult to perform inference over BNN's as the marginal distribution cannot be computed analytically, and we have to resort to approximations such as variational inference to optimize for variational lower bound [Graves, 2011, Blundell et al., 2015, Hernández-Lobato et al., 2016, Gal and Ghahramani, 2015].

## 6 Conclusions

In this work we developed an uncertainty-aware framework to improve self-training mechanism by exploiting uncertainty estimates of the underlying neural network. We particularly focused on better sample selection from the unlabeled pool based on posterior entropy and confident learning to emphasize on low variance samples for self-training. As application, we focused on task-specific fine-tuning of pre-trained language models with few labels for text classification on five benchmark datasets. With only 20-30 labeled examples and large amounts of unlabeled data, our models perform close to fully supervised ones fine-tuned on thousands of labeled examples. While pre-trained language models are natural few-shot learners, we show their performance can be improved by up to 12% using uncertainty-aware self-training. Some interesting future work include extending these methods to structured learning tasks like semantic parsing, multi-lingual settings with low-resource languages, and more real-world scenarios involving noisy or out-of-domain transfer data.

## Broader Impact

In this work, we introduce a framework for self-training of neural language models with only a few labeled examples.

This work is likely to increase the progress of NLP applications and drive the development of general-purpose language systems especially for domains with limited resources. While it is not only expensive to acquire large amounts of labeled data for every task and language, in many cases, we cannot perform large-scale labeling due to access constraints from privacy and compliance concerns. The latter concerns are amplified when dealing with sensitive user data for various personalization and recommendation tasks. Our framework helps in this regard for the NLP systems to obtain state-of-the-art-performance while alleviating privacy concerns.

To this end, our framework can be used for applications in finance, legal, healthcare, retail and other domains where adoption of deep neural network may have been hindered due to lack of large-scale manual annotations on sensitive user data.

While our framework accelerates the progress of NLP, it also suffers from associated societal implications of automation ranging from job losses for workers who provide annotations as a service as well as for other industries relying on human labor. Additionally, it suffers from similar concerns as with the use of NLP models by malicious agents for propagating bias, misinformation and indulging in other nefarious activities.

However, many of these concerns can also be alleviated with our framework to develop better detection models and mitigation strategies with only a few representative examples of such intents.

## Footnotes

[1]Code is available at `http://aka.ms/UST`

[2]A sentence is translated to a foreign language followed by back-translation to the source language. Due to noise injected by Neural Machine Translation systems, back-translation is often a paraphrase of the original.

[3]Note that these models use different encoders and pre-training mechanisms.

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
