[Supplementary Material · ust_appendix.pdf]

# Appendix: Uncertainty-aware Self-training for Text Classification with Few Labels

## 1 Pseudo-code

---

**Algorithm 1:** Uncertainty-aware self-training pseudo-code.

---

Continue pre-training teacher language model on task-specific unlabeled data $D_u$ ;
Fine-tune model $f^W$ with parameters $W$ on task-specific small labeled data $D_l$ ;
**while** *not converged* **do**
    Randomly sample $S_u$ unlabeled examples from $D_u$ ;
    **for** $x \in S_u$ **do**
        **for** $t \leftarrow 1$ **to** $T$ **do**
            $W_t \sim Dropout(W)$ ;
            $y_t^* = softmax(f^{W_t}(x))$;
        **end**
        Compute predictive sample mean $E(y)$ and predictive sample variance $Var(y)$ with
         Equation 11 ;
        Compute BALD acquisition function with Equation 6 ;
    **end**
    Sample $R$ instances from $S_u$ employing sample selection with Equations 7 or 8 ;
    Pseudo-label $R$ sampled instances with model $f^W$ ;
    Re-train model on $R$ pseudo-labeled instances with Equation 12 and update parameters $W$ ;
**end**

---

**Teacher-student training:** In our experiments, we employ a single model for self-training. Essentially, we copy teacher model parameters to use as the student model and continue self-training. Although, some works initialize the student model from scratch.

**Sample size.** Ideally, we need to perform $T$ stochastic forward passes for each sample in the large unlabeled pool. However, this is too slow. For computational efficiency, at each self-training iteration, we select $S_u$ samples randomly from the unlabeled set, and then select $R \in S_u$ samples from therein based on uncertainty using several stochastic forward passes.

## 2 Hyper-parameters

We do not perform any hyper-parameter tuning for different datasets and use the same set of hyper-parameters as shown in Table 1.

Also, we retain parameters from original BERT implementation from `https://github.com/google-research/bert`.

Table 1: Hyper-parameters

| Dataset | Sequence Length |
|---------|-----------------|
| SST-2 | 32 |
| AG News | 80 |
| DBpedia | 90 |
| Elec | 128 |
| IMDB | 256 |

| | |
|---|---|
| Sample size for selecting $S_u$ samples from unlabeled pool for forward passes in each self-training iteration | 16384 |
| Sample size for selecting $R$ samples from $S_u$ for each self-training iteration | 4096 |
| Batch size for fine-tuning base model on small labeled data | 4 |
| Batch size for self-training on $R$ selected samples | 32 |
| T | 30 |
| Softmax dropout | 0.5 |
| BERT attention dropout | 0.3 |
| BERT hidden dropout | 0.3 |
| BERT output hidden size $h$ | 768 |
| Epochs for fine-tuning base model on labeled data | 50 |
| Epochs for self-training model on unlabeled data | 25 |
| Iterations for self-training | 25 |

Table 2: UDA batch size.

| Dataset | Batch size |
|---------|------------|
| SST-2 | 32 |
| AG News | 32 |
| DBpedia | 32 |
| Elec | 16 |
| IMDB | 8 |

**UDA configuration.** Similar to all other models, we add validation data to UDA to select the best model parameters based on validation loss. We retain all UDA hyper-parameters from `https://github.com/google-research/uda`. We use the same sequence length for every task as in our models and select the batch size as in Table 2.

Note that our UDA results are worse than that reported in the original implementation due to different sequence length and batch sizes for hardware constraints. We select the maximum batch size permissible by the V100 gpu memory constraints given the sequence length.

**Code and data.** We have included the code for UST with README containing all the configurations and hyper-parameters. We have also uploaded[1] the data_directory and pre-trained BERT language model checkpoints used in our model.

## Footnotes

[1]`https://drive.google.com/drive/folders/1KzUdbRzBh3gzPx-HoIGalTBC20Sz9x6J?usp=sharing`