[Reviews · NeurIPS 2020]

Review 1

Summary and Contributions: This paper proposes a self-training framework that selects more effective training instances by computing uncertainty estimates, then selecting new samples based on model confidence. The proposed system outperforms other semi-supervised learning methods on a set of text classification benchmark datasets.

Strengths: * The work addresses a core challenge in machine learning -- how to train from fewer labeled examples. * The proposed framework offers an effective reinterpretation of ideas in semi-supervised learning. * The empirical evaluation is comprehensive. The results suggest that the proposed work generally offers improvements over comparative baselines.

Weaknesses: * While labeling training instances is a time consuming process, and labeling 30 instances is certainly preferable over labeling 30,000 instances, the bigger hurdle for many challenging NLP tasks is over developing a sound annotation scheme in the first place. In practical terms, a reduction of labels from hundreds to tens is nice but may pale in comparison to the overhead of the annotation schema development. * While the reported results show an improvement over comparative baselines, most of the improvements are rather modest.

Correctness: The methodology seem reasonable.

Clarity: The paper presentation is sufficiently clear.

Relation to Prior Work: The relation to prior work is clearly discussed.

Reproducibility: Yes

Additional Feedback: Figure 2 shows UST accuracy with K train labels/class. I assume that if I look at where each curve intersects K=30, I should get the results reported in Table 2. This does not seem to be the case for AG News (Graph shows not quite 84, Table reports 87.74) and SST (Graph looks to be between 86-87, Table reports 88.19). Could you explain this discrepancy? *** post author feedback: thank you for the explanation.


Review 2

Summary and Contributions: This paper proposes an uncertainty-aware self-training framework for text classification with only few labels, where the authors treat the model as Bayesian neural networks (BNNs) to characterize uncertainty and develop a new self-training scheme in such context with concept/methods borrowed from BNNs. The main techniques include: (a) use dropout to perturb weight in the context of Bayesian neural network to obtain pseudo labels; (b) use the BALD metric to measure uncertainty and design sample selection strategies based on it; (c) use confident training that enables the student model to focus more on confident examples. Combination of these techniques demonstrate strong performance on several text classification benchmarks with only 30 labels. ---------- After Rebuttal --------- Thank the authors for the response and it addresses most of my concerns, thus I would like to increase my score to 6.

Strengths: (1) Borrowing concepts/methods from Bayesian neural networks into modern self-training is novel. BNNs may be a better tool to characterize uncertainty than the normally used confidence score, and the authors develop a new self-training scheme within the BNN framework facilitating new sample selection strategies and confident training (2) Experimental results are competitive, and the proposed method demonstrates much smaller variance than other baselines

Weaknesses: My main concerns are on the experiments. While the authors make effort to perform ablation analysis, I think there are still some important missing ablations to convince me that such BNN-powerd self-training scheme is better than classic ST: (1) The proposed method always uses smart sample selection strategy while the classic ST baseline in this paper does not select samples or just select them uniformly. It is very common for classic ST to select samples based on confidence scores, which can be class-dependent as well. Thus I feel that the comparison made with classic ST is not very fair. I would like to see the comparison between UST removing Conf and classic ST with confidence-based and class-dependent sample selection, or just replace the sample selection part in full UST with confidence-score-based selection to see what happens, otherwise I don’t see any direct evidence to show that the BNN-powered “uncertainty-awareness” is better than simple confidence-score-based baseline. (2) Low variance displayed in Table 2 is a nice advantage of UST, but it is not very clear to me why the variance gets reduced. Does sample selection or confident training have a major effect on the variance? If so, does classic ST with confidence-based selection also have small variance? And how would the variance change if the confident training part is removed from UST? (3) The UDA numbers on IMDB are much lower than that reported in the UDA paper which is a bit concerning to me. I think the authors should include the reported numbers in the UDA paper in Table 2 as well and clarify what could be the reason for the performance gap in the main content instead of appendix, otherwise it is kinda misleading for readers who are not familiar with the UDA paper results or these benchmarks.

Correctness: Yes

Clarity: The paper writing can be better by fixing many typos

Relation to Prior Work: Yes

Reproducibility: Yes

Additional Feedback: line 97: parenthesis not closed Eq 6: the \hat and superscript t symbols overlap line 145: parenthesis incorrect Eq 12: Is this equation a typo? I feel it looks different from what is said in the text


Review 3

Summary and Contributions: The authors propose to take uncertainty into account in self-training. The intuition is that self-training can benefit from using samples that the teacher model is uncertain about, since it would lead to little improvements if the teacher model is already confident about a sample. The authors measure uncertainty by the entropy of predictions under different dropout samples. The proposed methods lead to improved performances on top of self-training and UDA.

Strengths: The paper is well-motivated. Sample selection in self-training is an important problem. The authors tackle the uncertainty measure problem in a principled way. The proposed methods lead to significant improvements on several text classification tasks. The paper is also clear, well-written and easy-to-understand.

Weaknesses: The empirical evaluation would be even stronger if more datasets are considered.

Correctness: Yes.

Clarity: Yes.

Relation to Prior Work: Yes.

Reproducibility: Yes

Additional Feedback: I have read the authors' responses and have no further questions.


Review 4

Summary and Contributions: This paper introduces a new semi-supervised learning algorithm---based on the classic self-training technique---for text classification with few labelled instances. The primary technical innovation of this paper is twofold: (i) a sampling strategy that takes into account the model's approximated uncertainty (estimated through the Monte Carlo Dropout technique of Gal and Ghahramani, 2016), and (ii) a loss function that takes into account the *variance* of the model's predictions (Eq. 12), where the model incurs a higher loss for misclassifying instances that has a lower variance under the Monte Carlo dropout (i.e. these are instances that the teacher model is fairly confident about). Experiments on five text classification benchmarks indicate that, in the low-resource scenario where only few labelled data are available, the approach substantially outperforms: (i) the standard BERT baseline + fine-tuning, and (ii) the "vanilla" self-training approach that does not take into account the model's uncertainty estimates into account. The approach also compares favourably to other strong semi-supervised learning baselines, such as the recently proposed Unsupervised Data Augmentation (UDA; Xie et al., 2019) that leverages consistency training and additional resources (e.g. backtranslation). -----After authors' response----- Thank you for the clarification. After reading the other reviews and the authors' response (which addresses most of my concerns), I maintain my initial assessment that this is a good paper. Hence I am keeping my overall score of "7".

Strengths: 1. The question of how we can design NLP models that can perform well with only few labelled instances---above and beyond the improvements we get from language modelling pretraining---is a really important research question. For instance, there are many low-resource languages and/or specialised domains (e.g. medical reports) where large amounts of labelled data are expensive or infeasible to collect; this paper takes a step towards building models that perform well under such limited data scenario. 2. The paper features strong empirical results that confirm the efficacy of the proposed approach, outperforming: (i) the standard BERT + fine-tuning baseline, (ii) a "vanilla" self-training approach, and (iii) other strong semi-supervised learning baselines, including UDA that leverages external resources like backtranslation. Table 4 also suggests that the proposed technique outperforms other methods that use more labelled instances per class. 3. The paper features fairly extensive ablation studies showing: (i) that both the uncertainty-weighted sampling procedure and the variance weighting on the loss are important (Table 3), and (ii) how the accuracy of the approach changes with more labels and self-training iterations. 4. The idea of using uncertainty estimates to improve self-training is an interesting one. The approach is also fairly theoretically grounded, since it relies on the uncertainty estimation procedure of Gal and Ghahramani (2015).

Weaknesses: 1. My main concern about this submission is that it presumes the existence of *in-domain* unlabelled training set that comes from the same distribution as the labelled instances. This is because the paper uses a large training set (e.g. IMDB classification), and then split that into: (i) the labelled training set (only a small fraction of the training data belongs in this category), and (ii) the unlabelled set (the rest of the training data is put here, where the true label information is discarded). This crucially guarantees that the examples in the unlabelled and labelled training sets are similar (i.e. they come from the same distribution/data-generation process) to one another. However, this presumption often does not hold in real practical setups: we may not have large amounts of in-domain unlabelled text readily available, or at least we have to try and find in-domain unlabelled data using some approximate similarity metric (which may be a noisy process on its own). I am not holding this point too much against this paper, since prior work follows the same pattern, but it would be good to apply the approach on a more realistic, *truly low-resource* setup, rather than a *simulated* low-resource setup as used in this work. 2. Some aspects of the clarity can be improved, as detailed in the "Clarity" and "Additional feedback, comments, and suggestions" section below.

Correctness: The methods proposed in this paper are theoretically sound---in particular, they are mostly based on the established prior work of Gal and Ghahramani (2015). The claims are also empirically well-validated by the set of experiments, which properly compare with the relevant baselines and semi-supervised learning prior work. The work also features some additional ablation studies to better understand where the gains are coming from.

Clarity: The paper is mostly well-written, although I have some questions and suggestions for improvements below: 1. In Eq. 12, $log p(y | f^{W} (x_u) . Var(y))$ should be $-log p(y | f^{W} (x_u) . Var(y))$ (i.e. missing a minus at the front). 2. Could you clarify further the term $\sigma^2$ in Eqs. (10) and (11)? How is this estimated? 3. In line 102, "in practise" should be "in practice". 4. Line 126 states that the model uses "hard pseudo-labels", which in my understanding means the single most likely label according to the multiple different dropout masks of the self-training teacher, and treating that as the "predicted ground-truth" label. How does this affect Eq. (1)? Does this mean that the target distribution is to predict 1 for that predicted label, and 0 everywhere else?

Relation to Prior Work: The work mostly compares with the relevant prior work on: (i) semi-supervised learning, especially for text classification (ii) uncertainty estimation in neural networks, and (iii) sample selection based on some criteria (e.g. curriculum learning). The differences between the proposed approach and some related work (e.g. UDA) are also clearly outlined.

Reproducibility: Yes

Additional Feedback: Some questions, comments, and suggestions: 1. The best approach seems to sample unlabelled text based on their "easiness" (i.e. samples that the teacher model is least uncertain about is more likely to be used), while the alternative of sampling unlabelled text based on their "hardness" seems to perform worse (Table 3). Is there a way of doing a combination of the two? E.g. for each batch, sample half of the instances based on the "easiness" criterion, and sample the rest based on the "hardness" criterion? This might strike a balance between just choosing the relatively simple examples that the model is confident about, and choosing the hard examples that the teacher is not very confident about. 2. Why does having more labelled instances not seem to help much for SST, Elect, and DBPedia? Figure 2 suggests that the model performance is almost the same for some of these tasks, even when comparing using only 20 labelled instances + the proposed approach, or all (many thousands) labelled instances?

[Author Response · NeurIPS 2020]

We thank all the reviewers for their time and insightful feedback about our work. We address one of the core challenges in training machine learning models with limited labels. This is crucially important for tasks with sensitive user data where we cannot manually access and annotate a lot of data, as well as for low-resource tasks in different languages. Many of the recent few-shot learning works focus on computer vision compared to NLU tasks. Despite the promise of pre-trained language models in overcoming the annotation bottleneck, we still see a gap in performance when these models are trained on a few samples (say, 20-30 samples in our setting) in contrast to thousands of annotations with an average accuracy gap of **14%** for tasks in our work. We leverage self-training with several advances to bridge this gap.

**R1 (Q1)** raises an important point with respect to developing a sound annotation scheme. Note that even if we resolve this challenge, it is still too expensive and in some cases infeasible to obtain large-scale human annotations for many specialized domains especially dealing with sensitive data. **(Q2)** From Table 2, we observe our method to be **4.2%** better than classic self-training and **2.2%** better than UDA. Note that UDA has access to a Neural Machine Translation system that generates paraphrases for consistency learning, whereas our model does not leverage any such external resource, and, therefore, is more general. Table 4 compares different models with the same setup as ours leveraging different forms of pre-training. We observe our model to obtain atleast **7%** improvement in IMDB and **4%** improvement in AG News over our closest baseline in the form of variational pre-training [Gururangan et al., 2019] and reinforcement learning with adverarial training [Li and Ye, 2018], while using **3x-6x less** training labels (shown by **K** in **Table 4**). **(Q3)** Table 2 reports the accuracy numbers averaged over runs with *multiple* random seeds for fair comparison across different models. While in Figure 2, we *fix* the random seed (for demonstration) and change other parameters within our model to show variations with different number of training labels and self-training accuracy over iterations.

**R2 (Q1)** The reviewer raises a good point regarding a simpler selection strategy that can be used as baseline with classic ST. Similar baselines reported for active learning [Gal et al., 2017] and preference learning [Houlsby et al., 2011] show the BALD measure outperforming them. Noting the above concern, we experimented with classic ST with confidence-based and class-dependent sample selection (as suggested by the reviewer) where confidence is given by predicted class probabilities. Preliminary experiments (over several runs with different seeds) show classic ST with such selection strategy to perform marginally better ($0.5\%$ acc. improvement with some task-specific variance) than classic ST (without selection) on an average in the few-shot setting (we will report detailed numbers in paper). Classic ST performs unbiased sample selection with uniform sampling forming a competitive baseline (often ignored in many works). Confidence-based sample selection (ignoring uncertainty) relies on the most confident predictions from a *weak* teacher resulting in early drifts from noisy pseudo-labels [Zhang et al., 2017]. This results from our few-shot setting where the teacher is fine-tuned on few labeled samples to start with, in contrast to many works employing such strategies with a stronger teacher. **(Q2)** Confident learning (Equation 11) incorporates sample variance (modeled by **Var(y)**) with minimization objective for *explicit* reduction. *Implicit* variance reduction happens via selecting samples with low uncertainty for self-training. Classic ST cannot use sample variance without using model uncertainty (that we achieve using MC dropout). Earlier baseline in (*Q1*) can be used to derive sample mean, but not the variance without accessing historical behavior or information from several stochastic passes. **(Q3)** raises an important concern regarding UDA. Consider the following differences. (1) Publicly available UDA code does not use validation set, instead, reports the maximum (across all epochs) and the last epoch accuracy on test set. We report UDA results on test set from the model with the best validation accuracy. (2) Recent works on data augmentation like SimCLR [Chen et al., 2020], UDA [Xie et al., 2019] and self-training with noisy student [Xie et al., 2020] show these techniques to work best with large batch sizes *as also applicable to our model*. Additionally, for IMDB longer sequence length plays a big role. For a fair comparison, with access to same amount of computational resources, we report UDA results and ours on the same hardware (V100 GPU) with maximum permissible batch-size and sequence length for every model. Due to page limitations, these settings were discussed in Appendix (lines 15-21) and will be moved to the main table as per suggestion. **(Q4)** As per our description in lines 186-189, Equation 12 should read $log(Var(y))$ (we will fix this typo).

**R3** Thanks for the feedback and suggestions. We are extending this work for more real-world tasks including multilingual settings where large-scale human annotations are difficult to obtain.

**R4 (Q1)** raises an important point regarding our simulation of the few-shot setting with in-domain unlabeled data (as also used in prior work). Extending these models to more realistic low-resource tasks with proxy/noisy data from related domains is an exciting direction for future work. **(Q2)** As per our description in lines 186-189, Equation 12 should read as $log(Var(y))$. Negative signs for cross-entropy loss and log inverse cancel out. **(Q3)** We do not need to estimate $\sigma$ for the minimization objective since it is independent of $y$. **(Q4)** Hard pseudo-labels are optimized with cross-entropy loss similar to hard ground-truth labels. **(Q5)** Sample mixing based on *easy* and *hard* examples is an interesting idea. We explored something similar with mixing equal number of instances sampled with BALD measure and remaining with uniform sampling. This presented mixed results that performed marginally better than ours for some of the runs with different seeds – warranting further exploration. **(Q6)** For tasks like DBpedia and Elec with very high performance given few training labels, there is diminishing returns on injecting more labels. In contrast, we improve more for tasks that are comparatively difficult like IMDB (very long reviews), AG News (4-class) and SST (very short snippets).

[Meta-Review · NeurIPS 2020]

This work presents a novel approach of integrating uncertainty into self-training to obtain strong results on text classification with very few labels. The work compares against a strong set of baselines and has extensive ablations. The reviewers agreed the response answered most of their concerns. The work could be improved with more diverse low-resource setups and by improving the clarity of the writing.